# Security Service Function Chain Based on Graph Neural Network

Wei Li [1], Haomin Wang [1], Xiaoliang Zhang [1,*], Dingding Li [2], Lijing Yan [2], Qi Fan [1], Yuan Jiang [1] and Ruoyu Yao [1]

[1] School of Control and Computer Engineering, North China Electric Power University, No. 2 Beinong Road, Changping District, Beijing 102206, China; liwei@ncepu.edu.cn (W.L.); whm@ncepu.edu.cn (H.W.); fq@ncepu.edu.cn (Q.F.); ginger@ncepu.edu.cn (Y.J.); yry@ncepu.edu.cn (R.Y.)

[2] The State Grid Henan Information & Communication Company, Zhengzhou 450052, China; fjogdbf@gmail.com (D.L.); fjifgkjdvkjfvjf@gmail.com (L.Y.)

* Correspondence: zhanghino@ncepu.edu.cn

**Abstract:** With the rapid development and wide application of cloud computing, security protection in cloud environment has become an urgent problem to be solved. However, traditional security service equipment is closely coupled with the network topology, so it is difficult to upgrade and expand the security service, which cannot change with the change of network application security requirements. Building a security service function chain (SSFC) makes the deployment of security service functions more dynamic and scalable. Based on a software defined network (SDN) and network function virtualization (NFV) environment, this paper proposes a solution to the particularity optimization algorithm of network topology feature extraction using graph neural network. The experimental results show that, compared with the shortest path, greedy algorithm and hybrid bee colony algorithm, the average success rate of the graph neural network algorithm in the construction of the security service function chain is more than 90%, far more than other algorithms, and far less than other algorithms in construction time. It effectively reduces the end-to-end delay and increases the network throughput.

**Keywords:** security service function chain; software defined network; network function virtualization; graph neural network

## 1. Introduction

With the continuous development of network services, people rely more and more on network services, and the requirements for network performance are also increasing. The traditional Internet architecture is difficult to dynamically deal with, given modern network applications. Moreover, the existing network services and applications need reliable and efficient servers [1], which makes the network services need to rely on expensive proprietary hardware equipment. At the same time, the function deployment is also seriously limited by geographical location, which not only affects the user experience, but also increases the consumption cost of space and energy [2].

On the other hand, with the rapid development of security business, attack methods are emerging one after another. Network security is also the focus of network services and network applications. The implementation of a network security service requires several security middleware devices to form a security service chain to defend against network attacks, which requires a large number of specific hardware environments to achieve specific security protection functions, and needs to be deployed in fixed locations in the network, which also leads to the consumption of a large number of network resources, cumbersome security service upgrading, expansion difficulties and other problems. Therefore, the background of cyberspace tells us that security protection cannot keep up with the pace of contemporary network security services. How to effectively and flexibly realize the security protection function has become a challenge of network security.

As a new generation network technology, software defined network (SDN) [3] and network functions virtualization (NFV) [4] can not only solve different network problems and meet business needs from different angles through their own level, but also closely combine to realize flexible network scheduling, dynamic expansion and rapid delivery on demand, generating greater value and meeting users' requirements for security business deployment to the greatest extent. SDN separates the control layer from the forwarding equipment through the separation of control and forwarding so that the network has the ability basis for software to flexibly define the network. Network administrators can easily define security control policies based on network flow and apply these security policies to various network devices so as to realize the security control of the whole network communication. NFV technology virtualizes the functions deployed by telecom operators on traditional proprietary hardware devices into virtual network functions (VNFs) [5] so that VNFs can replace proprietary hardware devices to provide services for users. While greatly reducing the operation cost, the functions get rid of the constraints of hardware devices, make their deployment and use more flexible, and realize the decoupling of functions and devices [6].

In recent years, graph neural network (GNN) [7] has become more and more widely used in social networks, knowledge maps, recommendation systems and even life sciences. The powerful function of graph neural network in modeling the dependency relationship between graph nodes has made a breakthrough in the research field related to graph analysis. Because each node in the network topology has its own structure information and characteristic information [8], graph neural network can also be well applied to the topology and has a good effect in building the security service function chain.

Quality of service (QoS) has traditionally been defined as throughput, end-to-end delivery delay, etc., while improving quality of service means ensuring throughput, reducing end-to-end delivery delay, and reducing data delay jitter. Requirements take into account metrics such as throughput, transmission latency, and availability. QoS is indispensable for the implementation of NFV architecture. Even after virtualization of network functions, the establishment of security service function chain, the mapping of forwarding maps and scheduling of network functions will be important factors for NFVI to consider in network service deployment. SSFC is the ultimate target of NFV-enabled networks [9].

This paper proposes a construction algorithm of a security service function chain based on graph neural network, and uses the method of graph neural network to design the construction scheme of security service function chain in a SDN/NFV environment.

The main contributions of this work are as follows:

- We propose a construction algorithm of security service function chain based on graph neural network. The algorithm uses the representation of nodes in graph neural network to construct a flexible and efficient security service function chain more comprehensively under the influence of its surrounding neighbor nodes.
- For the actual experiment, we use the Mininet network simulation tool and Floodlight software as the controller to simulate the real network.
- We test several most advanced artificial intelligence algorithms in generating the security service function chain. We evaluate our proposed model from the aspects of quality of service (end-to-end network delay and throughput) and security service chain construction time. Our proposed method has the best performance.

The structure of the rest of the paper is as follows: Section 2 introduces the related technologies; Section 3 introduces our proposed model; Section 4 introduces the evaluation of our proposed model; and finally, Section 5 summarizes the paper and puts forward the future work.

## 2. Related Work

This section mainly introduces the latest research on security service function chain in recent years, as well as the research progress on SDN/NFV applied by graph neural network.

Early in [10], a flexible and configurable dynamic composition mechanism of security service chain is proposed. This mechanism establishes a combined model based on vector space and integer programming, which reduces the transmission delay but increases the new reconstruction operation time overhead. Similarly, the author of [11] used integer programming to build a security service path model. Its construction mechanism considers the specific security requirements and the underlying resource state, but it is also easy to cause the security instances in the same security service chain to be deployed on multiple server nodes, increasing the link delay. The author of [12] only proposed the method of creating the security service function chain, without considering the consumption of network resources and delay. The author of [13] proposed a domain adaptation hybrid genetic algorithm for security service chain scheduling. Although it has a good improvement in bandwidth consumption and request success rate, it also does not consider the delay problem. Moreover, most of the methods proposed in the literature need to build the security service function chain according to the needs of users, which is far from being flexible and efficient. The author of [14] realized the automatic construction of security service chain according to security requirements, but it only considers the transmission delay. The author of [15] constructed the containerized security service function chain to reduce the network delay and reduce the resource consumption of placing the security service function chain. However, the proposed algorithm does not take into account the delay of different VNFs and limits of the type of network topology.

The graph neural network was first applied to the SDN /NFV field by the author of [16], who constructed a supervised learning method for service function chain traffic prediction using GNN. The author of [17] also used GNN to predict the resource demand of NFV, which approximated the performance of integer linear programming model in polynomial time and improved the reconfiguration overhead and resource utilization of service provision, respectively. The author of [18] used the graph neural network to predict various network performance indexes under a given routing strategy in the SDN network and achieved good results. Graph neural network is not only applied to the resource allocation of the service function chain because the establishment of service function chain needs to obtain the output network topology in order, so the sequence model in the graph neural network usually plays a substantive role in solving this problem. The author of [19] used the graph neural network sequence model to find an effective path to process all the requested virtualized network functions in order. The author of [20] used graph neural network to extract nodes and link resources to solve the VNF placement problem.

Through the above research, it is found that most works in the literature do not fully consider the constraints in the construction of the security service function chain, and most of them use an integer programming model, which takes a long time to draw a conclusion. It is easy to obtain the local optimal solution by using heuristic algorithms, such as the genetic algorithm. The related literature using graph neural network shows that its performance and prediction results have been well improved, but most of the literature is mainly combined with the service function chain, and there is no in-depth research on the construction of the security service chain. Therefore, combined with the popular graph neural network framework in recent years, a new optimization construction scheme of the security service function chain model is proposed.

## 3. Model Introduction

This section introduces the related technologies of the security service function chain based on graph neural network.

### 3.1. SDN and NFV

As a new network architecture, SDN realizes the separation of the control plane and forwarding. Through the open South interface, the SDN controller realizes the centralized control of the underlying forwarding equipment [21]; various network applications can make the network interface software open and programmable through the open North

programmable interface, which greatly improves the flexibility of network deployment and service deployment [22]. SDN divides the network into three parts: application layer, control layer and forwarding layer. The control layer is responsible for the generation and distribution of all forwarding logic. The author of [23] presented a machine learning powered SDN work. Therefore, this paper implements the model layer on the control plane of the software defined network. Figure 1 shows the basic architecture of SDN, in which the graph neural network model proposed in this paper is located in the control plane. The application layer controls the forwarding rules of flows in the network through the programmable interface provided by the control layer, while the infrastructure layer is responsible for carrying specific services [24].

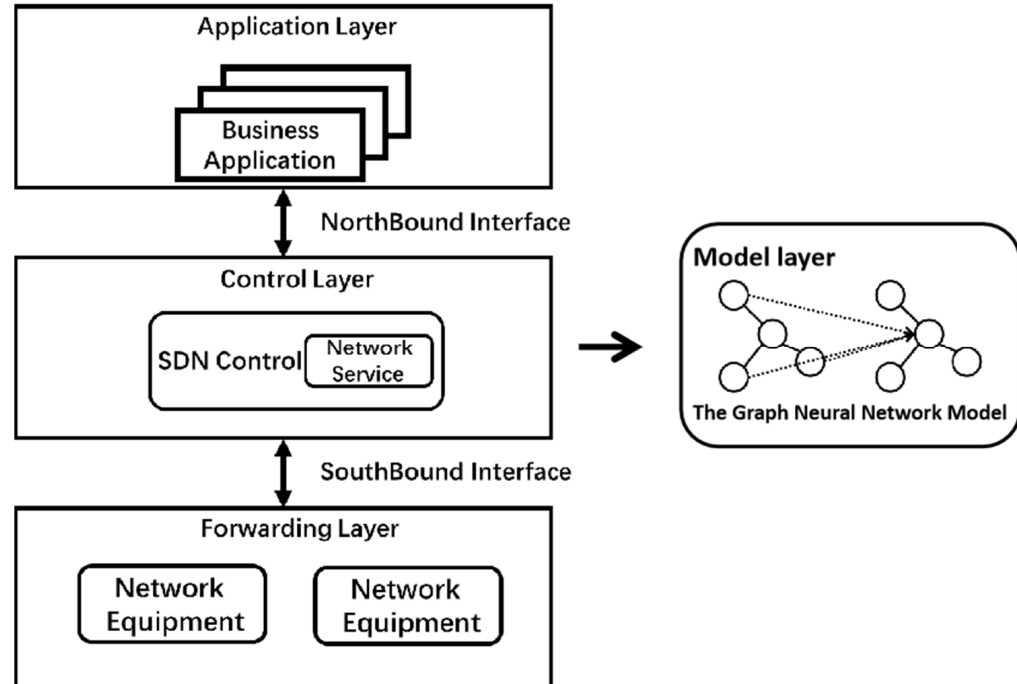

**Figure 1.** SDN architecture diagram with the graph neural network model layer.

NFV, network functions virtualization, deploys traditional communications technology services to the cloud platform (cloud platform refers to the virtual machine platform formed by virtualization of physical hardware, which can host communications technology and information technology applications) so as to realize the decoupling of software and hardware of 5G network elements [25]. NFV architecture mainly includes three parts: virtualized network functions, network function virtualization facilities, and network function virtualization management and arrangement [26].

The virtualized network function is software running on virtualized resources and deployed in virtualized or non-virtualized networks in order to realize network function. The network function virtualization facility includes a variety of hardware resources that can be virtualized, such as computing, storage and network resources. In addition, it also includes the unique RF antenna resources in the centralized access network architecture. The virtualization layer completes the abstraction of hardware resources, supports the execution of computing, storage and network connection functions, logically divides the resources and provides them to VNF for use so as to decouple the software from the underlying hardware. Network function virtualization management, scheduling management, the scheduling of hardware resources, a virtual resource layer, and virtualized network elements, and the scheduling and life cycle of complete network functions achieve the effects of high performance, high reliability and automation [27].

NFV and SDN come from the same technical basis [28]. SDN and NFV are independent of each other and do not necessarily depend on each other, but they can promote the flexibility of each other's deployment. The effective combination of SDN and NFV will reduce the dependence on network physical components that improve the security prospect [29]. Its framework is shown in Figure 2.

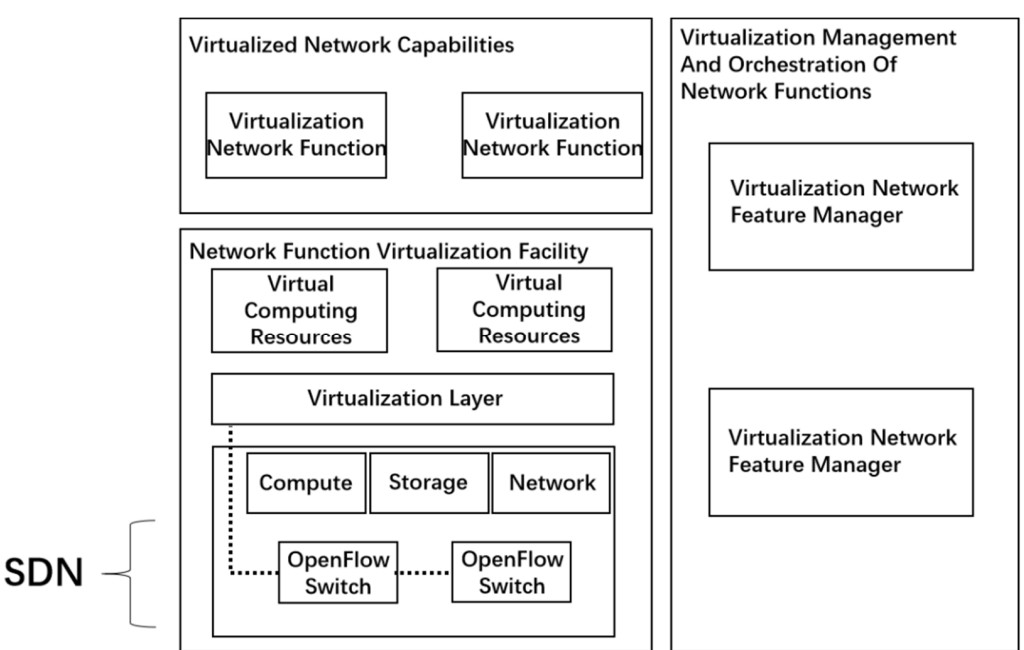

**Figure 2.** Contact architecture diagram of NFV and SDN.

*3.2. Graph Neural Network*

Graph and network structures exist in a large number of areas of life, such as urban rail transit networks, social networks, network topologies, etc. In computers, we collectively call this the data structure graph. The graph structure consists of each node *V* in graph *G* and edge *E* connecting each node.

$$G = (V, E) \tag{1}$$

The graph neural network is a kind of neural network running directly on graph structure [30], which automatically learns the characteristic information and structure information of the graph at the same time. The graph neural network is usually composed of two modules: propagation module and output module.

3.2.1. Propagation Module

The nodes in the figure transfer information and update status, including aggregator and updater.

Aggregator: its purpose is to learn the embedded representation $h_n$ of node *n* by aggregating the information of nodes around node *n*. Specifically,

$$h_n = f\left(x_n, x_{co[n]}, h_{ne[n]}, x_{ne[n]}\right) \tag{2}$$

where $x_{co[n]}$ . represents the feature of the edge connected to *n*, $h_{ne[n]}$ indicates the embedding of *n* adjacent nodes, and $x_{ne[n]}$ represents the characteristics of *n* adjacent nodes. *f* can be interpreted as the feedforward fully connected neural network.

Updater: the embedded representation of node n is updated iteratively in the learning process of the model.

$$H^{t+1} = F\left(H^t, X\right) \tag{3}$$

where $t$ is the $t$-th iteration, $X$ is all features, and $H^t$ is the embedded representation of all nodes in the $t$-th iteration.

### 3.2.2. Output Module

By setting the status $h_n^T$, characteristic $x_n$ is passed to the output function $g$ to calculate the output of GNN. The model outputs the label corresponding to each node after the last iteration:

$$o_n = g\left(h_n^T, x_n\right) \tag{4}$$

Among them, $g$ can be interpreted as the feedforward fully connected neural network. At this time, the loss function of the model is

$$\text{loss} = \sum_{i=1}^{p} (t_i - o_i) \tag{5}$$

where $t_i$ is the real label of the $i$th node, $o_i$ is the output label of the model, and $p$ is the number of nodes.

### 3.3. Security Service Function Chain

The service function chain is a collection of ordered service functions. It formulates different service function paths to process data according to policies. Based on the service function chain, the security service function chain (SSFC) is mainly responsible for the security service function and the specific processing of the received messages, such as the firewall, IDS, IPS and other security devices [31]. It is a security service function and defines an orderly set of virtual security functions.

The security service function chain and network topology of the traditional network are closely coupled, and the deployment is complex. When the service chain is changed and expanded, the network topology needs to be changed, and the network equipment needs to be configured again [32]. Driven by the development of SDN and NFV technology, the security service function chain forms a complete network security application architecture, which can provide flexible and controllable network security services for the existing network environment, and its modularity and integrity are very mature. It can not only inherit the traditional security service functions, but also make new improvements for the virtualized environment so as to improve the expansibility, stability and security [33]. The security service function chain realizes user isolation, personalized customization and other functions in detail, and derives a unique architecture. The system flow chart of the security service function chain [34] is shown in Figure 3.

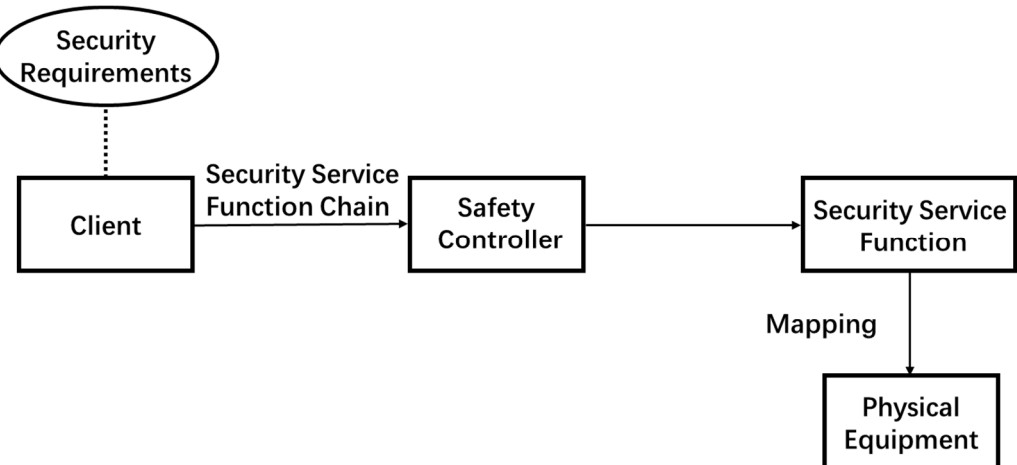

**Figure 3.** Overall system flow chart of security service function chain.

Like any SFC, SSFC is guaranteed by QoS, including performance, reliability and availability. When considering vnf-enabled networks, the placement of each vnf instance constituting SSFC will directly affect these requirements. Therefore, in order to provide end-to-end services, multiple vnf instances of different types should be accessed in a specific order to create SSFC. Firstly, the security service function chain is generated according to the corresponding security requirements, and the information is sent to the security controller. The controller generates the security service function path according to the security service function chain, passes through the security service function in sequence, and realizes the forwarding path mapping from the logical SSFC to the physical device. The system logic representation of security service function chain is shown in Figure 4.

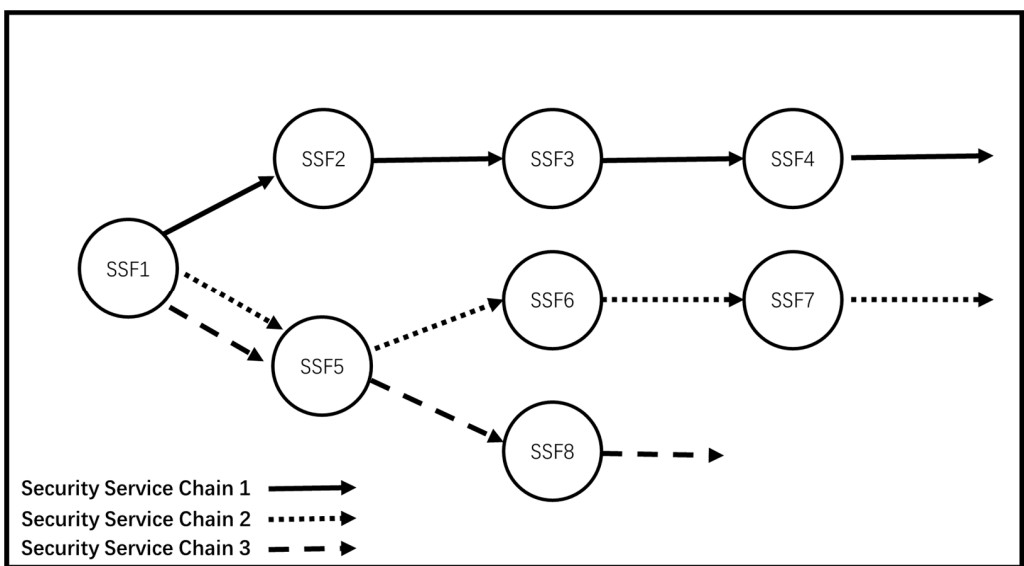

**Figure 4.** System logic diagram of security service function chain.

Each circle represents a different security service function through a network. The arrows represent three different security service function chains, which contain a set of specific security service functions connected in order.

*3.4. Security Service Function Chain Based on Graph Neural Network*

By iteratively calculating the characteristic information and structure information of link nodes in network topology, the problem of constructing a security service function chain for the network topology is transformed into a real-time prediction problem of link nodes based on graph neural network, considering the guarantee of QoS. The construction process of the security service function chain is shown in Figure 5.

Because graph neural network is a neural network that operates graph data, the same method can be used to build a security service function chain from network topology. For the input of the model, the characteristics of each node in the nodes of the network topology are represented as a characteristic matrix of $N \times F$ and a matrix with dimension $N \times N$ of the graph structure, where n is the number of nodes in the graph network and F is the input characteristic number of each node. Using the state vector $h\_n$ represents the state of node n in the network topology. The state of node n needs to be calculated with four parts of vectors: the eigenvector of node n, the eigenvector of the neighbor node, the state vector of the neighbor node and the eigenvector of edge (connected with n). We change the location of non-existent neighbor nodes to null values.

Then we carry out iterative calculation in the graph neural network. We use the Relu activation function and dropout to alleviate the problems of gradient disappearance and overfitting in the iterative calculation of each layer in the process of learning the characteristics of link nodes in the graph neural network. Finally, the softmax function is

used as the activation function to obtain the prediction node corresponding to the maximum probability in the probability matrix and construct the security service function chain. The graph neural network model is shown in Figure 6.

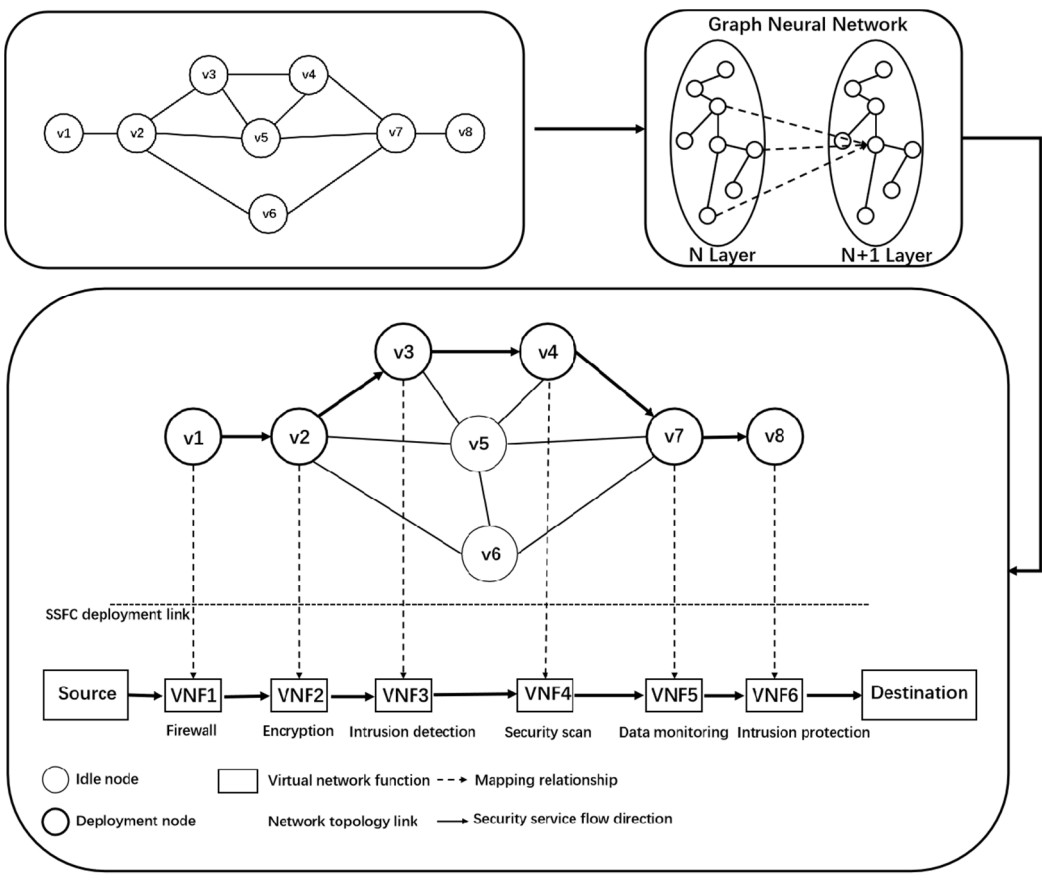

**Figure 5.** Construction process of security service function chain.

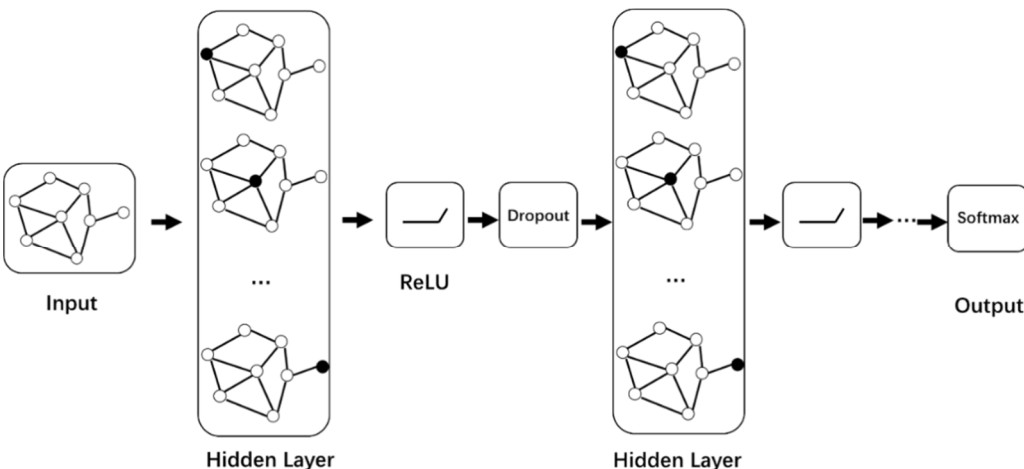

**Figure 6.** Architecture of graph neural network model.

## 4. Results and Evaluation

This section mainly introduces the experimental results of using graph neural network to construct the security service function chain, and compares our proposed method with the construction methods in recent years: greedy algorithm, shortest path algorithm, bee colony algorithm and other related traditional algorithms. The proposed model is compared

with them in terms of the network delay, throughput, construction time and success rate of the security service function chain.

In this paper, we use the Mininet lightweight simulation tool Mininet [35], the open source Floodlight controller to issue traffic forwarding rules, and the Mininet topology generator to simulate the real network topology. The Floodlight Openflow Controller and Openflow Switch uses Openflow for protocol switching in Mininet. The Mininet simulation network is composed of a virtual host, OVS switch, controller and link. The network analysis tools are Wireshark and iPerf [36]. Then, according to various characteristic training, a security service function chain with the best performance is constructed in sequence through Python code.

### 4.1. Experimental Environment

Our experimental operations were conducted on a Linux 64-bit operating system, an Intel Xeon E5-2680 processor, and an NVIDIA GeForce RTX 3080 with 64 GB of memory. The experimental environment language was Python, and the deep learning framework was TensorFlow.

### 4.2. Data Settings

In this paper, 10 different network functions are defined and numbered as 1~10. Each security service request is composed of one or more of them. The constraints of the experiment are also re-elaborated here. Each node in the network topology can carry any VNF, and it is set to arrive at a security service every five time units; each service function monopolizes the resources of one node until the processing is completed. Other setting conditions are shown in Table 1 and better simulate the real environment to build a real and effective security service function chain. In this experiment, we set the link bandwidth of the underlying network to a large number and the bandwidth required by the virtual link to a random number between 5 and 10 so as to ignore the blocking factor of the link. The parameters in the experiment are randomly selected from the maximum and minimum values under the condition of uniform distribution. Details are shown in Table 1.

**Table 1.** Experimental parameter range.

| Parameter | Minimum | Maximum |
|---|---|---|
| Number of nodes/pieces | 5 | 50 |
| Number of processing functions of a single node/piece | 1 | 5 |
| Single function processing time/MS | 100 | 500 |
| Length of security service chain/piece | 3 | 10 |
| Bandwidth required for virtual link | 5 | 10 |

We adjust the training parameters for many times and analyze the loss value to obtain the appropriate training parameters of the graph neural network model: we keep other parameters unchanged and reduce the learning rate $\alpha$ from 0.01 to 0.5; when the learning rate is 0.05, $\text{loss}(y, \hat{y})$ is the smallest. Similarly, the optimal dropout coefficient is 0.1, the L2 regular term is 0.002, the network parameters $W_0, W_1$ are (0,1), and the number of hidden layer nodes $L_{\text{node}}$ is 32.

### 4.3. Experimental Tests and Results

In order to verify that the performance of our proposed model is the best, we compare the model with the existing greedy algorithm, hybrid bee colony algorithm and shortest path algorithm. The experimental results are shown in Figures 7 and 8.

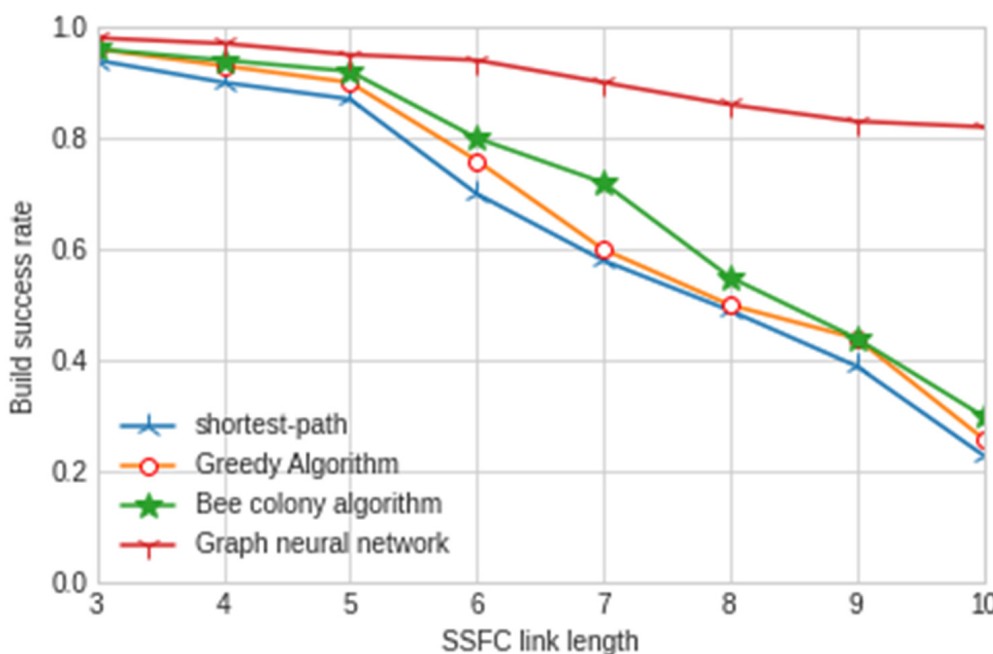

**Figure 7.** Construction success rate of different SSFC lengths.

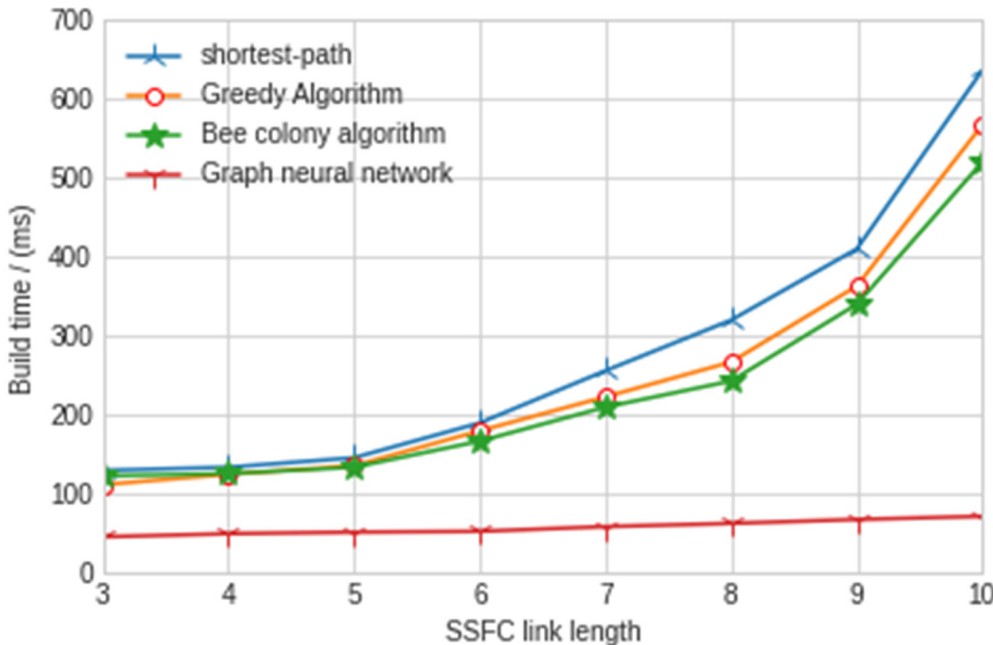

**Figure 8.** Construction time of different SSFC lengths.

According to the above experimental results, we can obviously find that the construction success rate of the security service function chain under different SSFC lengths is more than 80%, while the other three traditional artificial intelligence algorithms used in the mainstream are only less than 40%; thus, the success rate is greatly improved. For the construction time, the model in this paper is only 76 ms, which is far lower than the other three traditional artificial intelligence algorithms. It can better reflect the performance and stability of the graph neural network algorithm used in this model and can provide security services for users faster. In order to more deeply reflect the advantages of the graph neural network algorithm in building the security service function chain, we carried out relevant experiments on traffic intensity, network delay, jitter and throughput. The experimental results are shown in Figures 9–11 below.

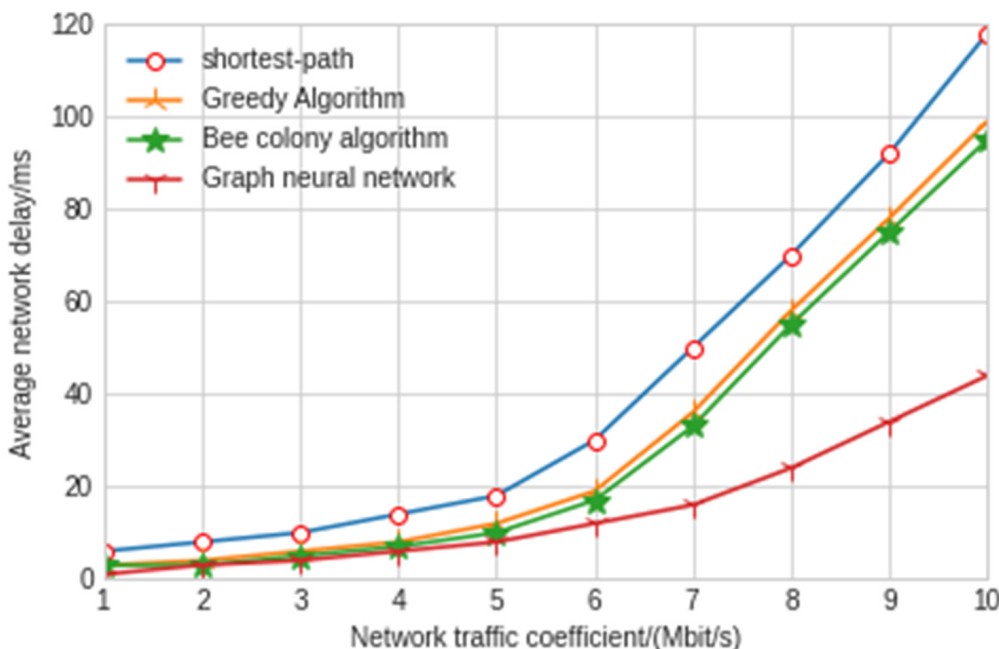

**Figure 9.** Average end-to-end network delay of different algorithms.

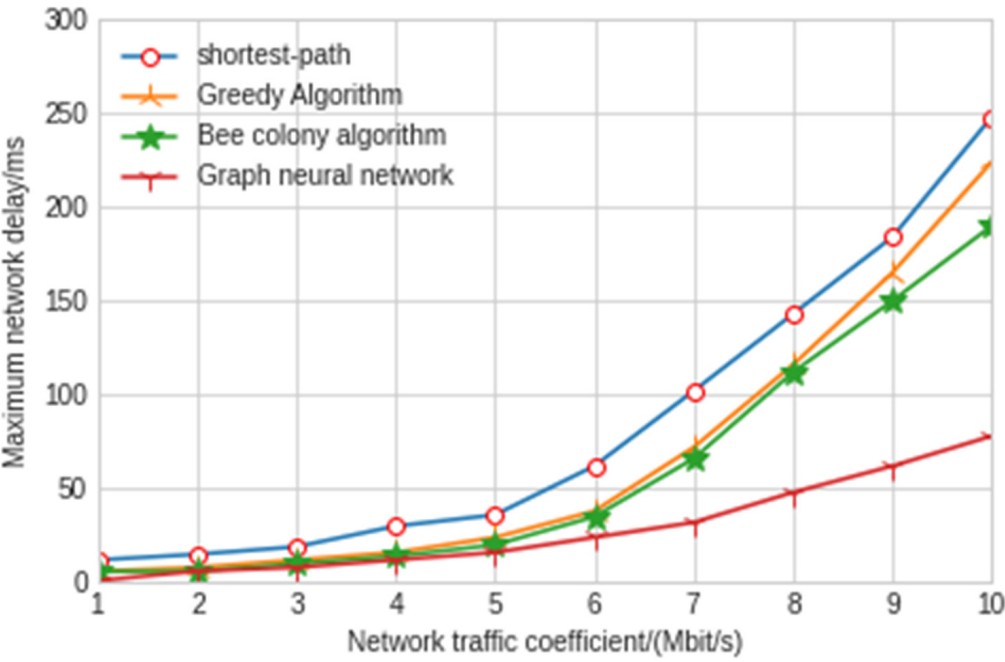

**Figure 10.** Maximum delay of end-to-end networks with different algorithms.

The experimental results show that the shortest path algorithm is the worst in terms of throughput and network delay. The maximum delay and average delay of the graph neural network model in this paper are 77 ms and 43 ms, respectively, which is nearly twice shorter than the other three artificial intelligence algorithms. As can be seen from Figure 11, the throughput of the model experiment in this paper is 2.97 Mbit/s, which is also far lower than the other three algorithms. The greedy algorithm and hybrid bee colony algorithm can achieve relatively close results. Because the greedy algorithm uses local information when making an action decision, the greedy algorithm has slightly greater jitter than the bee colony algorithm. The results also show that the graph neural network algorithm improves a lot compared with the other three algorithms, and the delay and throughput are far better than the other three algorithms. Moreover, in the network security environment

with high delay requirements, the delay of control signaling in the SDN controller and switch cannot be ignored. At this time, the graph neural network algorithm has very significant advantages. With the increase in traffic intensity, the performance of the graph neural network is better and more stable. The graph neural network algorithm can make decisions with less observation information after training through the surrounding node information, which reduces the interaction delay with the controller. In addition, when the SDN controller fails, the graph neural network algorithm can also maintain good results.

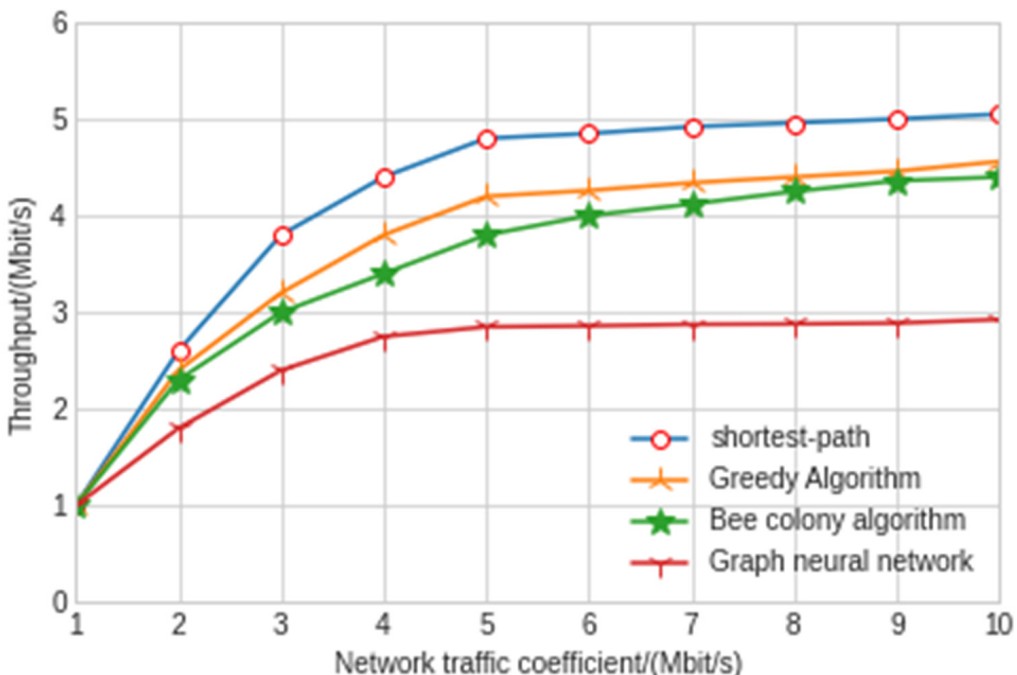

**Figure 11.** Throughput comparison of different algorithms.

## 5. Conclusions and Future Work

Traditional methods simplify the network model, ignore many key security information in the network, and usually cannot well predict the network performance index so as to build a security service function chain with better performance. In the SDN environment, this paper proposes a security service function chain construction model based on the graph neural network. The model can access multiple VNF instances of different types in a specific order to create SSFC. It can capture the complex relationship between physical links and routing policy paths in the network topology so as to quickly and accurately predict the corresponding QoS performance indicators, such as delay and throughput under the given topology, routing policy and traffic matrix so as to establish the security service function chain efficiently. However, the current mainstream routing optimization algorithms are usually based on heuristic algorithms. The establishment of security service function chain takes a long time, with low efficiency and high possibility of a local optimal solution. Compared with the current mainstream routing optimization algorithms, our model not only outperforms other model algorithms in QoS, but also makes great progress in establishment time. In the future, if we build the security service function chain based on the prediction model proposed in this paper and combined with other artificial intelligence algorithms, there will be a more efficient solution.

**Author Contributions:** Conceptualization, W.L. and H.W.; methodology, X.Z.; software, D.L.; validation, L.Y., W.L. and H.W.; formal analysis, Y.J.; investigation, Q.F.; resources, R.Y.; data curation, H.W.; writing—original draft preparation, W.L. All authors have read and agreed to the published version of the manuscript.

**Funding:** This work was financially supported by the science and technology project of State Grid Henan Electric Power Company: "Research and application of key technologies of adaptive protection for State Grid Cloud Security" (Grand No. SGHAXT00YJJS2100034).

**Institutional Review Board Statement:** Not applicable.

**Informed Consent Statement:** Not applicable.

**Data Availability Statement:** Not applicable.

**Conflicts of Interest:** The authors declare no conflict of interest.

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
