# Peer review of "Security Service Function Chain Based on Graph Neural Network"

_information, doi:10.3390/info13020078_

Round 1
Reviewer 1 Report
The paper explores an intriguing topic, which leverages Graph Neural Network (GNN) to construct Security Service Function Chain. The paper presents detailed technical backgrounds and designs as well as mininet-based simulations results.
However, I have a couple of concerns and questions about the paper:
First, the motivation of the paper is not clearly described in the paper. What is the key insight why GNN can help the construction of a security service chain? performance or functionality? the paper should clearly present its unique merits and demonstrate them in the design and evaluations, like [19] focused on focusing on QoS of service function chain.
Second, the comparison in the paper is not convincing. I did not figure out the unique contributions of the proposed schemes compared with [19]. This should be aligned with the first point, the author should emphasize the unique contributions of the paper instead of simply using GNN for the service function chain.
Third, some contents of the paper are not in good shape and should be greatly polished. For example, most of the figures in the paper are blurred; table 1 is oversized; Figure 3 is not related to the design.
Reviewer 2 Report
The paper present and interesting work on SDN/NFV and Neural networks.
The figure 1 need to be enhanced in order to show the architecture build where the controller is located and how the part of neural networks is utlised in the paper. An interesting work that can be used is "Towards a Machine Learning Based Situational Awareness Framework for Cybersecurity: An SDN Implementation " where the authors present a machine learning powered SDN work.
Figure 2 is too simplified not showing any interaction of the system. The paper needs to work on the figure to make simple to follow the interaction of the subcomponents.
Line 164 to 167 I don't see a point on discussing in the specific section what is NFV.
Figure 3 is just for taking space nothing to learn as it is well known.
The section 3.3 most probably is written by different authors so it not following the rest of the paper.
The section 4 needs to be enhanced with further discussion of the results. The text is too short and does not clear present the findings.
Finally the paper needs to provide in a clear manner what is the contribution on the field and this can be done in section 5.
The reference needs to be enhances with work be done in Europe and USA like 10.1109/WF-IoT.2019.8767249, 10.1109/NETSOFT.2016.7502419 and 10.1109/GLOBECOM38437.2019.9013246
Round 2
Reviewer 2 Report
The authors addressed all my comments.